# A Microfluidic Approach for Assessing the Rheological Properties of Healthy Versus Thalassemic Red Blood Cells

**DOI:** 10.3390/mi16080957

**Published:** 2025-08-19

**Authors:** Hao Jiang, Xueying Li, Zhuoyan Liu, Siyu Luo, Junbin Huang, Chun Chen, Rui Chen, Fenfang Li

**Affiliations:** 1Department of Electrical and Electronic Engineering, Southern University of Science and Technology, Shenzhen 518055, China; 12232176@mail.sustech.edu.cn; 2Shenzhen Bay Laboratory, Institute of Biomedical Engineering, Shenzhen 518107, China; lixueying921@yeah.net (X.L.); liuzhuoy@szbl.ac.cn (Z.L.); m19852337009@163.com (S.L.); 3Pediatric Hematology Laboratory, Division of Hematology/Oncology, Department of Pediatrics, The Seventh Affiliated Hospital of Sun Yat-Sen University, Shenzhen 518107, China; huangjb37@mail.sysu.edu.cn (J.H.); chenchun@mail.sysu.edu.cn (C.C.)

**Keywords:** red blood cells, deformability assessment, shape recovery, microfluidic platform, thalassemia

## Abstract

The deformability of red blood cells (RBCs) is critical for microvascular circulation and is impaired in hematological disorders such as thalassemia, a prevalent public health concern in Guangdong, China. While microfluidics enable high-precision deformability assessment, current studies lack standardization in deformation metrics and rarely investigate post-deformation recovery dynamics. This study introduces an automated microfluidic platform for systematically evaluating RBC deformability in healthy and thalassemic individuals. A biomimetic chip featuring 4 µm, 8 µm, and 16 µm wide channels (7 µm in height) was designed to simulate capillary dimensions, with COMSOL CFD numerical modeling validating shear stress profiles. RBC suspensions (10^7^ cells/mL in DPBS) were hydrodynamically focused through constrictions while high-speed imaging (15,000 fps) captured deformation–recovery dynamics. Custom-built algorithms with deep-learning networks automated cell tracking, contour analysis, and multi-parametric quantification. Validation confirmed significantly reduced deformability in Paraformaldehyde (PFA)-treated RBCs compared to normal controls. Narrower channels and higher flow velocities amplified shear-induced deformations, with more deformable cells exhibiting faster post-constriction shape recovery. Crucially, the platform distinguished thalassemia patient-derived RBCs from healthy samples, revealing significantly lower deformability in diseased cells, particularly in 4 µm channels. These results establish a standardized, high-throughput framework for RBC mechanical characterization, uncovering previously unreported recovery dynamics and clinically relevant differences in deformability in thalassemia. The method’s diagnostic sensitivity highlights its translational potential for screening hematological disorders.

## 1. Introduction

Normal human red blood cells (RBCs) have a distinctive biconcave disc shape, measuring approximately 8 µm in diameter and about 2 µm in thickness. During microcirculation, they can undergo passive deformation while maintaining mechanical stability [1]. The deformability of RBCs depends on the structural properties of the cytoskeletal components [2], the vertical interactions between the cytoskeleton and the integral transmembrane complexes [3], as well as intracellular viscosity, RBC hydration state, and surface area–volume interactions [4]. Healthy RBCs readily respond to shear stress in the microcirculation, allowing them to deform effectively and pass through capillaries. Changes in the deformability of RBCs directly influence cell movement, morphological maintenance, and intercellular interactions [5], thereby affecting functions such as cell migration [6], mechanosensing [7], and signal transduction [8]. These alterations significantly influence the progression and severity of cardiovascular diseases, metabolic disorders, and neurological conditions [9,10]. The restoration of RBC morphology following deformation is essential for maintaining normal physiological function [11]. Mechanistically, this restorative process is governed by two intrinsic biophysical parameters: (i) the surface area-to-volume ratio (SA/V), which dictates the geometric constraints for reversible shape transitions [12], and (ii) the viscoelastic properties of the erythrocyte membrane, whose dynamic behavior is modulated by membrane-associated proteins [13]. Understanding both the deformability and shape recovery of RBCs is essential for elucidating their biophysical effects.

To measure the deformability of RBCs, several experimental methodologies have been proposed, which can be broadly categorized into direct and indirect methods. Direct methods include micropipette aspiration, optical tweezer, twisting micromagnetic beads, atomic force microscopy, parallel plate rheometry, and controlled cavitation rheology. Micropipette aspiration technology [14,15] employs a pressure difference to aspirate cells into a micropipette in a quasi-static way, allowing for the assessment of cellular deformability and viscoelasticity. Optical tweezer technology [16,17] and twisting micromagnetic beads [18] utilize laser beams and a magnetic field, respectively, to capture microbeads attached to the cell surface, generating a controlled force to assess cell mechanics [19]. Atomic force microscopy (AFM) [20,21] measures the mechanical properties of cells by obtaining force–distance curves at the cell surface, enabling high-precision elasticity measurements of individual living cells under physiological conditions. Parallel plate rheometry [22] quantifies the deformation of single cells under external stress. These methods assess the time-resolved response to force and extract parameters related to cellular deformability. However, these techniques are technically demanding and time-consuming, limiting measurement efficiency and making high-throughput measurements difficult.

In contrast, microfluidic-based methods [23] represent a highly attractive alternative due to their potential for single-cell precision and high-throughput deformability analysis [24,25,26]. The controlled cavitation rheology method stretches a collection of RBCs using impulsive flow generated by a single laser-induced cavitation bubble in a microchannel [27]. It can stretch many cells simultaneously and allows one to obtain large initial cell deformations and yield strength [28,29,30]. However, this direct method generates heterogeneous forces and cell deformations and requires integration of optics with microfluidics. In recent years, indirect methods for measuring the deformability of cells based on microfluidic technology have demonstrated ease of operation, low cost, and high throughput. They can be categorized into three types: constriction-based deformability cytometry (cDC), extensional flow deformability cytometry (xDC), and shear flow deformability cytometry (sDC) [31]. With the conventional cDC method, cells deform when passing through a narrow constriction channel with dimensions smaller than their diameter [32,33], where cell deformability is influenced by factors such as surface friction, stiffness, viscoelasticity, and adhesion properties. The Suspended Microchannel Resonator (SMR) quantifies cellular stiffness by measuring the time required for a cell to pass through a constriction channel comparable in size to the cell itself [34]. However, the measurement results provide a composite indicator of cellular deformability and do not characterize the deformation process of the cell within the channel or its recovery after exiting the constriction. The xDC methods quantify cellular stiffness by measuring the deformation ratio of the long axis to the short axis of cells in an extensional flow field, assuming an elliptical cell shape [35,36]. Multiparametric deformability cytometry (m-DC) [37] was developed based on this, integrating parameters such as cell size, deformability, deformation kinetics, and morphology. However, due to the anisotropic nature of RBCs [11], this method is not suitable for measuring their deformability.

The sDC method is a non-contact flow cytometry method that applies hydrodynamic flow shear to cells in a microchannel with dimensions slightly larger than the cell diameter at a constant flow velocity, inducing mechanical cell deformation under a strong shear stress gradient [38,39,40]. Recent studies have utilized high-speed charge-coupled device (CCD) cameras to capture real-time images of cells at various stages of deformation [41]. Additionally, some studies have integrated microscopic cell imaging with numerical simulations to further investigate the factors contributing to red blood cell deformation [42]. Moreover, due to their operational simplicity, these methods hold potential for applications in biological laboratories and clinical settings [43,44]. In recent years, the rapid advancement of microfluidic technology has provided an excellent platform for studying the biophysical properties of RBCs under both physiological and pathological conditions.

The deformability of RBCs is influenced by many pathological conditions, such as malaria [45], diabetes [46,47], sickle cell disease [48], thalassemia [49], hereditary spherocytosis [50], and hereditary xerocytosis [51]. The deformability of RBCs can be affected by metabolic processes that control ATP levels and redox state [52]. The inability to maintain deformability leads to a shortened RBC lifespan, and if not compensated for by the production of new RBCs, can result in hemolytic anemia [53]. Therefore, reliably estimating the deformability of RBCs and understanding the factors that control this process are crucial for assessing the severity of a patient’s condition and selecting the best treatment strategies. For example, thalassemia, a globally significant hereditary blood disorder, is closely associated with the biophysical properties of red blood cells. Caused by autosomal gene defects, the disease results in an imbalance in the synthesis ratio of α- and β-globin chains in hemoglobin [54,55]. The unpaired globin chains precipitate and adhere to the red blood cell membrane skeleton, reducing the membrane’s viscoelasticity and thereby impairing the deformability of red blood cells. This ultimately leads to hemolytic anemia [49].

In this study, we developed a high-throughput method for assessing RBC deformability based on microfluidic technology. The RBC suspension was introduced into a microfluidic channel with a microinjection pump. A high-speed camera was employed to record the complete motion of RBCs as they traversed a narrow channel at tens of thousands of frames per second (fps). A custom-developed program was utilized to analyze the deformation of RBCs throughout their motion, incorporating a deep learning algorithm to accurately extract RBC contours. Four parameters were established to characterize deformability. Additionally, we observed the morphological recovery of RBCs after exiting the narrow channel and compared several different RBC models, including healthy RBCs, RBCs fixed with paraformaldehyde (PFA), and those from patients with thalassemia. The results suggest that this method holds promise as a potential tool for disease diagnosis using label-free biophysical markers of RBCs.

## 2. Materials and Methods

### 2.1. Cell Sample Preparation

Whole blood samples (1–2 mL) were collected from the veins of healthy donors during medical checkups or from patients with thalassemia at the Seventh Affiliated Hospital of Sun Yat-Sen University. Heparin was used as an anticoagulant. Both healthy donors and patients with thalassemia were aged between 2 and 8 years old and resided in Guangdong province. The Seventh Affiliated Hospital of Sun Yat-Sen University’s Ethics Committee approved the collection of patient samples, and the study complied with institutional norms, the Declaration of Helsinki, and guardians’ informed consent. The collected blood samples were stored and transported at 4 °C to the Shenzhen Bay laboratory within 2 h of collection.

The fresh whole blood samples were washed 3 times and resuspended in 1X DPBS (Ca^2+^+, Mg^2+^+) after centrifugation at 1000 rpm for 3 min to achieve a final RBC concentration of approximately 10^7^ cells/mL (the concentration of red blood cells in whole blood is approximately 4.7–6.1 × 10^9^ cells/mL [56]).

For PFA treatment, the above samples were resuspended in 0.5% PFA in 1X DPBS (Ca^2+^+, Mg^2+^+) and incubated at room temperature for 30 min, after which the supernatants were discarded after centrifugation and the pellets resuspended in 1X DPBS (Ca^2+^+, Mg^2+^+) to achieve a final cell concentration of ~10^7^ cells/mL.

### 2.2. Microfluidic Device Preparation

The capillary vessel-mimicking microchannel was designed using AutoCAD 2021 software. We used standard soft lithography techniques to fabricate the patterned SU-8-based master mold, from which the polydimethylsiloxane (PDMS, Sylgard 184 Silicone Elastomer Kit (Dow Corning (Dowsil), Midland, MI, USA), 10:1 mix ratio, cured at 60 °C for 4 h) microchannel was cast. The microchannel was then bonded to a glass slide (50 × 75 mm^2^) immediately after 50 s of treatment in a plasma cleaner (Zepto one, Diener, Ebhausen, Germany).

### 2.3. Experimental Operation for Cell Deformation and Recovery

The microfluidic chip was placed on an inverted microscope (BDS500, CNoptec, Chongqing, China). A syringe pump (R462, KDS, Issy-les-Moulineaux, France) was used to introduce the cell suspensions into the microfluidic chip at a flow rate of 0.01 μL/min, and the RBCs were focused on the channel center by a sheath flow of 1 × PBS driven by another syringe pump (R462, RWD, Shenzen, China) at 0.02 μL/min. RBCs experienced elevated shear stress and became deformed upon entering the narrow channel, but started to recover their shape after exiting the channel. A high-speed camera (Nova S12, Photron, Tokyo, Japan) was used to capture the deformation and shape recovery of RBCs through a 63× oil immersion objective at a frame rate of 10,000 fps and exposure time of 10 μs to minimize motion blur. A custom-built script was employed for automatic cell tracking and contour analysis, from which the velocity and deformation of RBCs were extracted. The number of pixels per RBC is critical for tracking cells and analyzing cell contours. In our study, each red blood cell occupied approximately 510 pixels in total, which we found to be sufficient to ensure accuracy of cellular contour detection.

### 2.4. Numerical Simulation

Numerical simulation of the flow field in the microchannel was performed with a finite-element solver (COMSOL Multiphysics 6.0 trial, CFD module, COMSOL, Burlington, MA, USA). The geometry of the microchannel was imported into COMSOL from AutoCAD. Non-slip boundary conditions were employed, and the initial velocity at the inlets was calculated from experimental conditions. Though the flow demonstrated a Poiseuille profile, we were more interested in the velocity along the channel centerline in the flow direction (x) as we used sheath flow and focused RBCs to the channel center. Based on both experimental recordings and numerical simulations, the steep velocity gradients along the flow direction (x) and the channel centerline at the entrance and exit of the narrow channel expose RBCs to markedly higher shear stress, resulting in notable extensional deformation. The shear stress σ herein refers to that related to this velocity gradient.

### 2.5. Image Processing and Data Analysis

#### 2.5.1. Cell Tracking

Custom-built MATLAB 2024b and Python 3.0 scripts were used to analyze each image frame of recorded cell motion and deformation. Image preprocessing was performed to improve the image contrast and enhance the edge areas by image filtering and adjusting the grayscale histogram. Subsequently, the current image was processed by subtracting a pre-saved background image. Then, grayscale thresholding and binarization operations were employed to extract the cellular regions in the images, from which the approximate location and area occupied by the cell in the current frame can be determined. The motion state of the cellular region in the current image was predicted using the Kalman filter algorithm. The Hungarian algorithm was utilized to match the cellular regions across different image frames [57,58]. This approach facilitated the acquisition of the complete motion trajectories of individual cells within the microfluidic channel.

#### 2.5.2. Cell Contour Analysis Based on Deep Learning Methodologies

The extracted cellular regions were cropped from the original images. Then, the Canny algorithm combined with image erosion and dilation was applied to extract the cell contours [59], from which cell deformability was calculated. After exiting the narrow channels, due to imaging conditions, artefacts from a discernible white rim of the cells rendered the accurate extraction of cell contours using the Canny algorithm, which was based on image gradient operations. To solve this problem, a deep learning approach was employed with the open-source YOLOv8 convolutional neural network for the semantic segmentation of cell images [60,61], see Figure 1.

A dataset for red blood cell segmentation was constructed, comprising 60 images each for the narrow channels of three different widths that cells pass through. In this training set, half of the cells were normal healthy controls, and the other half were red blood cells from patients with thalassemia. LabelMe was used for data annotation. The pre-trained YOLOv8 deep learning network model was subsequently applied to the annotated dataset to perform model training. This was followed by testing and validation using 20 new cell images. Then, the trained network model was called in a MATLAB program to perform inference, yielding precise cell contours for subsequent analysis of cell deformation and recovery. The integration of traditional image processing techniques with contemporary deep learning methodologies enhanced the accuracy and reliability of cell contour extraction, providing a robust foundation for analyzing cell deformation and recovery features.

#### 2.5.3. Cell Deformation Analysis

RBCs have a disc shape that can be distorted under deformation. Both our studies and those of others found that RBCs travelling through a microfluidic constriction channel typically adopt a limited set of morphologies [62]. We restricted our analysis to cells exhibiting the parachute shape, which was the most common in our study.

We detected the cell contour accurately, obtained the projected cell area A and perimeter P, and used four definitions of cell deformation. The centroid of an individual red blood cell was extracted from the obtained cell contours, from which the moving speed of the cell can be calculated between image frames. We assume an ellipse fit to the cell contour for aspect ratio and Taylor deformation, using the major and minor axes a and b. For the non-circularity index and changes in the major axis, we did not fit the cell contour to an ellipse. Based on these obtained parameters, four distinct definitions of cell deformation were generated:

Aspect ratio:a/b

Taylor deformation:Ta = (a − b)/(a + b)

Non-circularity:$$D=1-2(\pi {A)}^{\frac{1}{2}}/P$$

Changes in the long axis of the cells:$$a/a_{0}$$where a_0_ is the initial long axis of the cell before entering the narrow stretching channel.

#### 2.5.4. Cell Shape Recovery Analysis

Aspect ratios *a/b* were extracted from cell contours after the cells exited the narrow channel. The dataset of *a*/*b* vs. time for each cell was first smoothed and then fitted with an exponential function *f* = *C*_1_
*+ C*_2_ *× exp(−t*/τ*)* with MATLAB using its built-in Non-Linear-Least-Squares and the robust method of fitting Bisquare and Trust-Region. Only data with goodness of fit (R-squared) above 0.95 were retained for subsequent statistical analysis of cell shape recovery.

The shear elastic moduli and viscous moduli for individual RBCs were estimated using methods described by Hocumuth et al. [63,64] and recently demonstrated in a similar experimental protocol to this study by Mancuso et al. [65]. We first use the classic Kelvin–Voigt model:(1)$$T=\frac{E}{2}\left(\lambda^{2}-\lambda^{-2}\right)+\frac{2\eta }{\lambda }\cdot \frac{\partial \lambda }{\partial t}$$
where *T* is the uniaxial tension force, *λ = a*/*a*_0_ is the stretch ratio, *E* is the elastic shear modulus, and *η* is the viscous modulus. The tension per unit length from purely uniaxial flow (valid along the centerline of our channels) is thus approximated as follows:(2)$$T=\frac{\sigma A}{L_{0}}=\frac{3A\mu \frac{du}{dx}}{L_{0}}$$
where *σ* = 3$\mu \frac{du}{dx}$ is the shear stress from a uniaxial extensional flow along the centerline of the channel, μ  is the suspending fluid viscosity, *A* = 136 μm^2^ is the average area of the RBC membrane, and *L*_0_ = 8 μm is the average resting length of a human RBC membrane. Settings (1) and (2) are equal to each other, and solving for $\frac{\partial \lambda }{\partial t}\;$ yields a first-order nonlinear ordinary differential equation for *λ*,(3)$$\frac{\partial \lambda }{\partial t}=\frac{1}{2\eta }\cdot (\frac{3A\mu \frac{du}{dx}}{L_{0}}\cdot \lambda -\frac{E}{2}(\lambda^{3}-\lambda^{-1}))$$

### 2.6. Statistical Analysis

If only two groups were compared, a two-tailed Student’s *t*-test was used for the statistical analysis of the deformability parameters. If more than two groups were compared, a one-way ANOVA test was performed first to determine whether one or more treatment groups were significantly different, followed by a post-hoc Tukey HSD multiple comparison test to identify which of the treatment pairs were significantly different from each other.

## 3. Results

### 3.1. Microfluidic Device Design, Experimental Setup, and Deformability Characterization

A microfluidic chip was designed to mimic small capillary vessels, where RBCs were stretched in a 200 μm long narrow part of the channel with widths of 4, 8, or 16 μm. Cell motion and deformation were imaged by a high-speed camera on an inverted microscope (Figure 2A). Each microfluidic unit had two inlets and one outlet. RBCs were focused by the sheath flow of 1× DPBS to the channel center before entering the narrow stretching region to ensure the reproducibility of the experiment (Figure 2A inset and Figure 2B). Numerical simulation of the flow field in the microchannel revealed the focusing effect of the sheath flow (dashed black box in Figure 2C) and the elevating velocity in the narrow stretching channel (red box in Figure 2C). The flow velocity in the narrow channel accelerated to around 0.07 m/s in the channel center, while the channel walls acted as a non-slip boundary, and the flow velocity before the entry and after the exit of the narrow channel varied between 0.01 and 0.02 m/s. These velocity gradients generated shear stress on cells entering and exiting the narrow channel, leading to cell deformation and shape recovery at relevant moments.

As shown in Figure 3A, an RBC was stretched when entering the narrow channel at 1.7 ms, it remained stretched from 3.0–5.0 ms until it exited the narrow channel at 6.3 ms and gradually recovered its shape (11.4 ms). Custom-built MATLAB and Python scripts were used for cell tracking (Figure 3B left). Traditional image processing techniques were integrated with contemporary deep learning methodologies to enhance the accuracy and reliability of cell contour extraction (Figure 3B right). Based on the obtained cell trajectories and cell contours, cell velocities and different definitions of cell deformation, including aspect ratio (*a/b*), Taylor deformation *Ta*, non-circularity, and change in the major axis of the cells (*a/a_0_*), were calculated. Figure 3C shows that both the cell velocity and deformation increased when the cell entered the narrow stretching channel, remained at a high level, and declined when the cell exited the narrow channel. It is interesting to note that the deformation reached its peak value at the entrance of the narrow channel (e.g., ~1.7 ms), which can be explained by the velocity gradient along the cell flow direction, e.g., larger flow velocity on the right side of the cell compared to the left side. Similarly, when the cell exited the narrow channel and encountered the velocity gradient, its elongated shape was compressed in the flow direction due to the larger flow velocity on the left side of the cell.

To validate the operation and measurements of the microfluidic devices, we performed a control experiment comparing the deformability of non-treated healthy RBCs vs. PFA-treated RBCs (Figure 4). Differential deformability was observed between the two groups based on microfluidic measurements. The PFA-treated cells have a significantly lower deformability among the four different definitions, consistent with the fixation and stiffening effects of PFA. These results suggest that our measurements and analysis of the cell deformability are valid.

### 3.2. Influence of Narrow Channel Width and Flow Velocities on Cell Deformation

Under physiological conditions, RBCs need to squeeze through small capillary vessels of various sizes for circulation, where they may experience different flow velocities, shear stress, and deformation; therefore, it is necessary to investigate the influence of narrow channel width and flow velocities on the deformation of RBCs. As shown in Figure 5A, cell deformability increased significantly when the narrow channel width decreased across all four definitions of cell deformability, while the average flow rate was kept constant at 0.03 μL/min. Similarly, significantly higher cell deformability was observed when the average flow velocity increased while the narrow channel widths were kept constant at 8 μm (Figure 5B). These results are consistent with numerical modeling, which reveals that the flow shear stress increases as a result of elevated velocity gradient in a smaller channel width or larger average flow velocity (Figure 5C–E). For example, the instant shear stress from the uniaxial extensional flow along the centerline of the channel is defined as $\sigma =3\mu \frac{du}{dx}$ [66] and the average flow shear stress along the flow direction at the entrance of the narrow channel and along the center line can be calculated as $\sigma_{ave}=3\mu \frac{∆u}{∆x}$, where μ is the dynamic viscosity of the medium (998 kg∙m^−3^) and $\frac{∆u}{∆x}$ is the average velocity gradient along the flow direction (along the centre axis). The average flow shear stress is estimated to be 12.0, 27.9, and 60.3 dynes/cm^2^. The max flow shear stress σ_max_ = 3μ (du/dx)_max_ along the center streamline is calculated from the maximum value of the first-order derivative of the position dependent velocity curves in Figure 5D, and estimated to be 25.8, 51.6, and 138.9 dynes/cm^2^ for narrow channel widths of 16, 8, and 4 μm at a constant flowrate of 0.03 μL/min (Figure 5F). The average flow shear stress is estimated to 2.61, 8.1, 12.9, and 17.7 dynes/cm^2^, and the max flow shear stress is estimated to be 10.2, 17.1, 27.0, and 36.9 dynes/cm^2^ for average flow velocities of 15, 25, 40, and 55 mm/s at a narrow channel width of 8 μm, respectively (Figure 5G). Therefore, we choose 8 μm narrow channel width and 55 mm/s average flow velocity for all subsequent measurements of cell deformability, as these conditions are within the physiological range (15 to 55 mm/s) [67], from which larger cell deformation can be obtained.

### 3.3. Label-Free Mechanical Phenotyping of RBCs from Patients with Thalassemia and Healthy Controls

Next, we employed the microfluidic approach to screen the deformability of RBCs in healthy controls and pathological conditions of thalassemia. Figure 6A shows the transit of an exemplary RBC from a patient with thalassemia through the narrow channel with the 8 μm width. The cell was stretched to an elongated shape when entering the narrow channel at 1.3 ms, it remained stretched from 1.3–4.6 ms until it exited the narrow channel at 5.3 ms and gradually recovered its shape (5.3–8.5 ms). Statistical analysis shows significantly lower cell deformability in the thalassemia group compared to the healthy control group, as observed using a narrow channel with widths of 4, 8, and 16 μm (Figure 6B–D). It is interesting to note that there is a more pronounced difference in aspect ratio between the two groups with smaller channel widths and larger shear stresses, indicating that the aspect ratio *a/b* may serve as a label-free biomarker to distinguish the deformability between RBCs from healthy donors and patients with thalassemia.

### 3.4. Shape Recovery of RBCs from Patients with Thalassemia and Healthy Controls

We further investigated the dynamics of cell shape recovery when RBCs exited the narrow stretching region (Figure 7A), comparing representative individual RBCs from healthy control (non-treated and PFA-treated) and thalassemia groups. RBCs in all three cases gradually recovered their shape as they moved from the channel exit (0–267 s). Compared to RBCs from healthy controls and non-treated samples, PFA-treated RBCs exhibited rapid recovery from 0 to 267 µs, while little change in shape was observed thereafter. In contrast, it took longer for non-treated RBCs from healthy controls to recover their shapes (0–2935 s). Interestingly, RBCs in the pathological thalassemia condition seemed to recover their shape even more slowly. The aspect ratio *a/b* can be described using the following equation:$$\gamma \left(t\right)=C1+C2{\cdot e}^{\left(-\right)t/\tau )}$$
where *C*1 and *C*2 are constants, τ denotes the characteristic relaxation time for cells to recover their shape, and *t* represents the time elapsed after the cells exit the narrow channel. The deformation data for each cell in each group are fitted using this equation (Figure 7B left) and a double logarithmic plot of (*γ*(*t*) − *C*1) vs. time reveals that the shape recovery follows an exponential decay (ln (*γ*(*t*) − *C*1) = ln(*C*2) − *t*/τ) (Figure 7B right), consistent with previous studies [64,68]. Statistical results demonstrate that RBCs treated with PFA exhibit the fastest recovery rate, while RBCs in the thalassemia group recover significantly slower than those from healthy controls.

## 4. Discussion

To investigate differences in deformability between RBCs from thalassemia patients and healthy donors, we established a microfluidic flow cytometry based on shear flow deformation and high-speed optical imaging. To characterize the dynamic behavior of multiple RBCs in microfluidic flow cytometry, we developed a custom image processing program that enabled precise tracking of individual moving RBCs and accurate extraction of cell contours. From the extracted RBC contours, we quantified cellular deformability with multiple definitions by calculating key morphological parameters. This approach successfully differentiated between RBCs from thalassemia patients and healthy donors. We further investigated recovery dynamics following the exit from constricted microchannels. Statistical analysis revealed significant differences in the recovery rates between healthy and thalassemic RBC models.

The microfluidic system developed in this study enables precise control over RBCs’ deformability and comprehensively records RBC motion before, during, and after transit through constricted microchannels. By implementing the sheath flow configuration, RBCs were hydrodynamically focused along the channel centerline, allowing cells to experience consistent flow shear with similar initial velocities when entering the stretching channel. To validate the reliability of our system, PFA-treated RBCs were compared with healthy RBCs, thereby establishing the platform’s sensitivity in detecting biomechanical alterations. Furthermore, through precise modulation of flow velocity, RBC deformation was achieved across varying channel widths, revealing the influence of the flow shear on cell deformation. Our microchannels (4–60 μm in width) closely mimic physiological vascular scales [69], providing biomimetic relevance to in vivo microcirculation. In our study, the Reynolds number *Re* < 0.72, indicating a laminar flow regime without turbulent disturbances [70], enhancing the stability and controllability of cellular deformation processes. The system design improved experimental reproducibility, establishing a robust tool for investigating RBC rheological properties.

In this study, we have substantially improved image analysis by integrating a Kalman filter and Hungarian algorithm. It allowed continuous tracking of multiple individual RBC trajectories throughout microfluidic recordings, compared to only one cell tracking in previous studies [36], which eased the requirement for sample concentration. This framework was combined with a YOLOv8 deep learning network to achieve high-precision RBC contour extraction [71]. A curated RBC image dataset was constructed to train the model, yielding both exceptional contour detection accuracy (>95% mean average precision) and computational efficiency. Conventional edge detection algorithms [59] suffer significant limitations under suboptimal imaging conditions, particularly when RBC boundaries exhibit low contrast against the background. The deep learning approach effectively circumvents this constraint by learning discriminative morphological features. Collectively, this integrated tool provides a robust analytical framework for quantitative investigations of RBC dynamics.

Furthermore, we defined a suite of parameters to quantify RBC deformability, including the aspect ratio (*a/b*), Taylor deformation (*Ta*), non-circularity, and relative elongation change (*a/a_0_*). These metrics collectively capture multidimensional aspects of cell morphological changes, enabling a comprehensive assessment of cellular deformability. While *a/b* and *Ta* directly reflect cellular elongation, the non-circularity characterizes contour irregularity. This multiparameter approach not only encompasses a broader spectrum of morphological features during deformation but also enhances assessment accuracy through integrated analysis.

Our results demonstrated significantly impaired deformability in thalassemic RBCs compared to healthy controls under the same experimental conditions. This deformation deficiency likely stems from structural and functional abnormalities in the RBC membrane [72]. Reduced deformability compromises microcirculatory perfusion efficiency, potentially exacerbating tissue hypoxia and triggering pathophysiological sequelae such as endothelial damage and inflammatory cascades [73,74].

To investigate recovery kinetics post deformation, we analyzed three RBC cohorts using our microfluidic platform: (1) healthy controls, (2) paraformaldehyde (PFA)-fixed RBCs (modeling rigidity), and (3) thalassemic RBCs. PFA-fixed RBCs exhibited the fastest post-constriction recovery, attributable to membrane stiffening and stabilized cytoskeletal architecture [75]. Conversely, thalassemic RBCs showed significantly slower recovery than healthy cells, consistent with disease-induced membrane defects and cytoskeletal dysfunction [76]. The elastic shear modulus and viscous modulus of RBCs could be estimated using Equations (1)–(3) described above. The elastic shear modulus E for the representative RBC from the control group is calculated to be more than three times bigger than that of the thalassemia groups (with their shape characteristic relaxation time *τ* at around the mean value of their respective groups). This finding is consistent with previous studies, which report that the average elastic shear modulus was higher for softer cells [63]. The viscous modulus *η* for the same representative RBC from the control group is found to be more than five times larger than that of the thalassemia groups, further confirming the disease-induced defects in cell structure and dysfunction in viscoelastic properties. These findings in our study elucidate distinct biomechanical signatures in strain recovery, providing mechanistic insights into cellular impairments associated with hematological disorders.

To approximate physiological conditions, the dimensions of the microfluidic constriction channel (7 µm in depth and 4, 8, 16 µm in width), the resulting RBC velocities (on the order of 10–100 mm/s) and shear stress for extensional deformation (1–10^2^ dynes/cm^2^) fall within the ranges reported for human blood vessels and flow speed of human erythrocytes [77,78]. However, several limitations exist. First, the microchannel in our study is rectangular in shape, unlike the cylindrical shape found in vivo. Therefore, RBCs were exposed to a more heterogeneous shear stress in the experiment compared to homogeneous shear stress in the radial direction in vivo. Nevertheless, we were more interested in the velocity gradient in the flow direction along the channel centerline and the resulting shear stress. Second, we used a highly diluted and RBC-only suspension in DPBS buffer prepared from whole blood. Thus, the extracellular environment differs markedly from native blood plasma, e.g., the fluid viscosity/density, chemical environment, and cell–cell communications could differ significantly [79]. Third, our experiment was performed at room temperature, which is lower than the in vivo temperature, and may affect the viscoelastic properties of RBCs and their deformability [80]. Fourth, our studies have a limited sample size (<150 cells), and considerable variability is observed in the measured parameters across the cell population. Due to the large field of view and high frame rate required to capture complete cell trajectories and deformations, tens of thousands of high-resolution images were generated within seconds. This resulted in a limitation on the number of cells that could be recorded per unit time, owing to the constrained storage capacity of our high-speed imaging system. The variability in the measured parameters could arise from the inherent heterogeneity within the RBC population [81]. Nevertheless, by employing sheath flow to focus cells toward the channel center and precisely detecting their contours, we reliably measured and distinguished statistically significant differences in cell deformability across groups.

Overall, the microfluidic system and data analysis pipeline presented serve as a robust platform for assessing the rheological properties of RBCs in health and disease, demonstrating good potential for in vitro screening of thalassemia and other hematological diseases.

## Figures and Tables

**Figure 1 micromachines-16-00957-f001:**
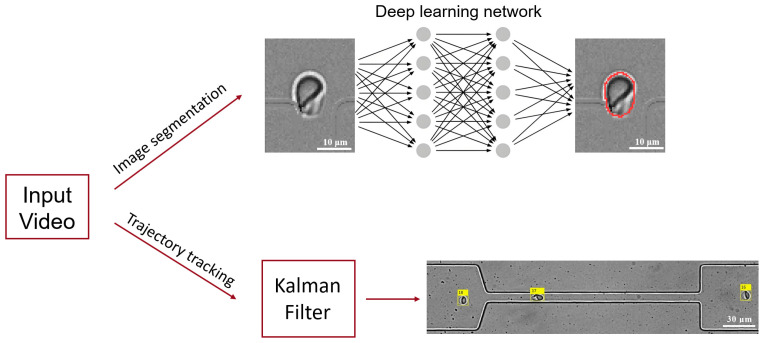
A schematic diagram of the core workflow for image processing of microfluidic experimental videos. It consists of two main steps: image segmentation and multi-object tracking. The YOLOv8 deep learning network model is employed to achieve accurate segmentation of the contour of red blood cells, and the Kalman filter algorithm is utilized to track the trajectories of multiple red blood cells in the video images. The red curve delineates the contour of the RBC, while the yellow square indicates the index of the RBC identified.

**Figure 2 micromachines-16-00957-f002:**
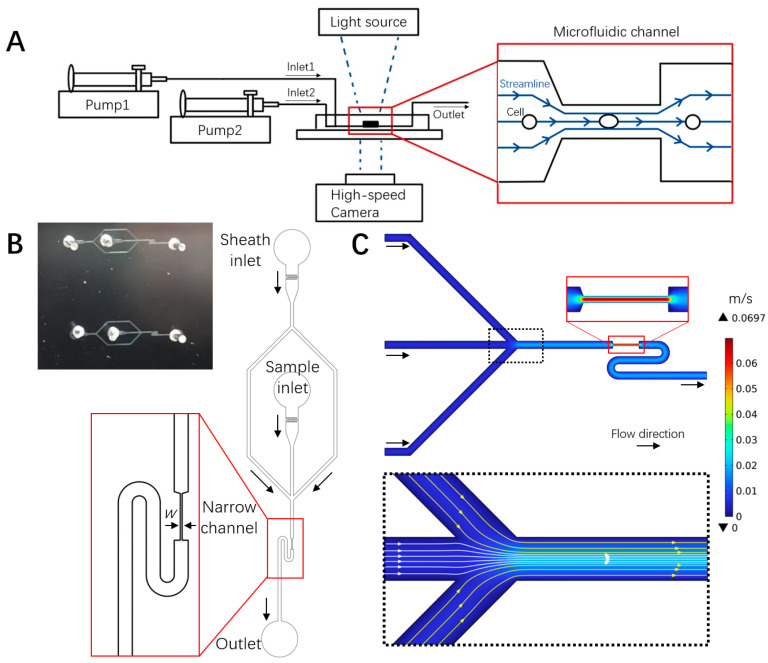
Microfluidic deformability cytometry of RBCs and flow field characterization. (**A**) Schematic diagram of the experimental setup, cell deformation, and flow field in the microfluidic chip. (**B**) Photograph (top left) and schematic of the microfluidic chip with sheath flow structure, where w is the narrow channel width (4, 8, 16 μm). The turning S shape following the exit of the narrow channel accommodated the size of the imaging field of the high-speed camera, allowing for the tracking of shape recovery. (**C**) Numerical simulation of the flow field in the microchannel.

**Figure 3 micromachines-16-00957-f003:**
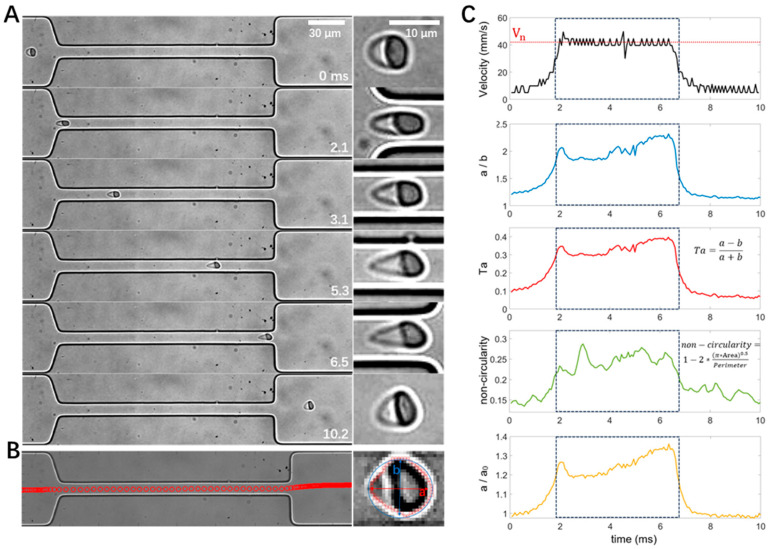
Illustration of cell motion, deformation, image processing, and data analysis with an example RBC from a normal healthy control. (**A**) Image sequence showing the deformation of the RBC passing through the narrow channel. (**B**) Cell trajectory and contour detection, with a and b depicting the major and minor axes, respectively. (**C**) Temporal evolution of cell velocity and deformability in different definitions measured from the same recording as in panel A. The dashed box denotes the time window when the cell was flowing inside the narrow channel. The width of the narrow channel is 16 μm.

**Figure 4 micromachines-16-00957-f004:**
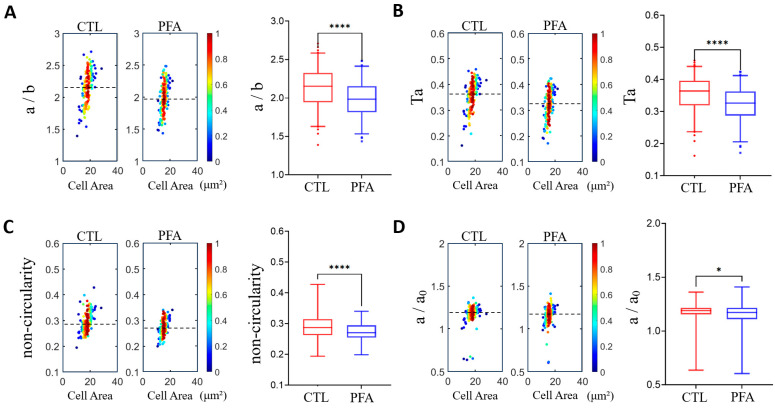
Validation of measurements by comparing the cell deformability of normal healthy RBCs and paraformaldehyde (PFA)-treated RBCs using density scatter plots and box plots. (**A**) Aspect ratio *a*/*b*. (**B**) Taylor deformation *Ta*. (**C**) Non-circularity. (**D**) Relative change in the major axis *a*/*a*_0_. The dashed lines indicate median deformability. A hotter color indicates a higher data density. * *p* < 0.05, **** *p* < 0.0001. The narrow channel width is 8 μm, with an average cell velocity of 100 mm/s. *n* = 165 for normal RBCs and 144 for PFA-treated RBCs.

**Figure 5 micromachines-16-00957-f005:**
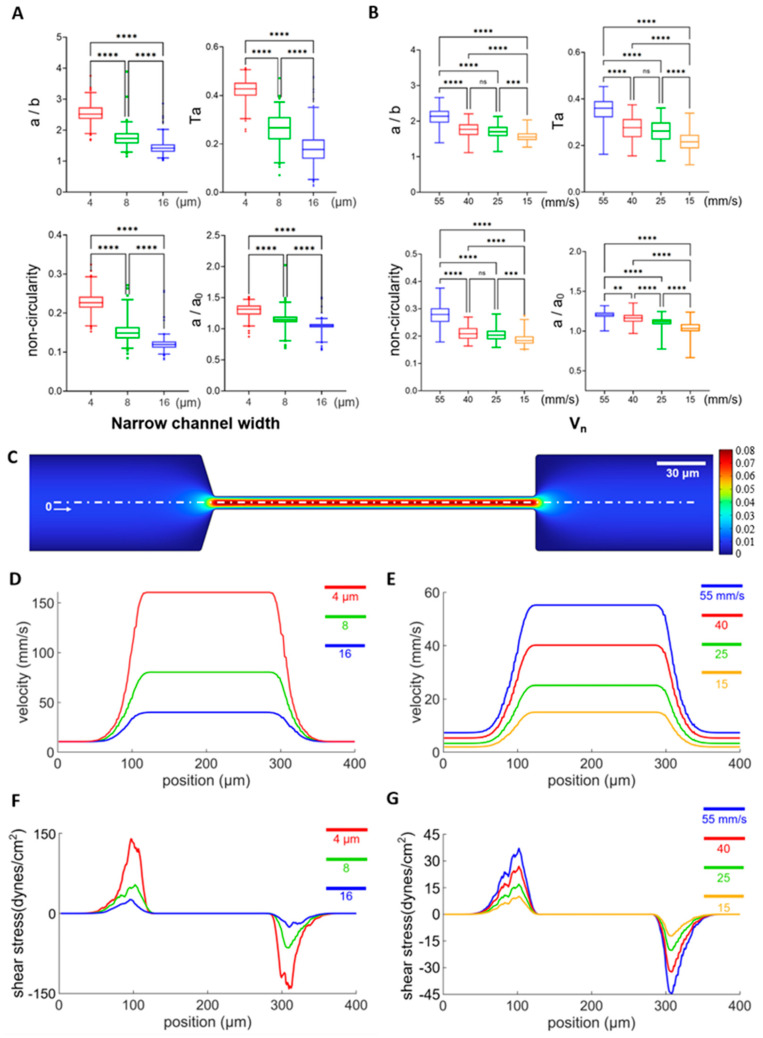
The influence of narrow channel width and flow rate on cell deformation. (**A**) Cell deformability from 4 definitions measured from narrow channel widths of 4, 8, 16 μm under a flow rate of 0.03 μL/min, and their statistical analysis. *n* = 182, 194, 265 for narrow channel widths of 4, 8, 16 μm, respectively. (**B**) Cell deformability from 4 definitions measured with maximum flow velocities (V_n_) of 55, 40, 25, 15 mm/s with a narrow channel width of 8 μm. The corresponding *n* values are 64, 98, 125, and 58, respectively. (**C**) Representative results of numerical modeling of the flow field across the arrow channel with a channel width of 8 μm and V_n_ of 40 mm/s. (**D**–**G**) Flow velocity or shear stress distribution along the white dashed line (x) in panel C at narrow channel widths of 4, 8, and 16 μm with the flow rate fixed at 0.03 μL/min or maximum flow velocities (V_n_) of 55, 40, 25, and 15 mm/s with the narrow channel width fixed at 8 μm. ** *p* < 0.01, *** *p* < 0.001, **** *p* < 0.0001, one-way ANOVA test followed by a post-hoc Tukey HSD multiple comparison test.

**Figure 6 micromachines-16-00957-f006:**
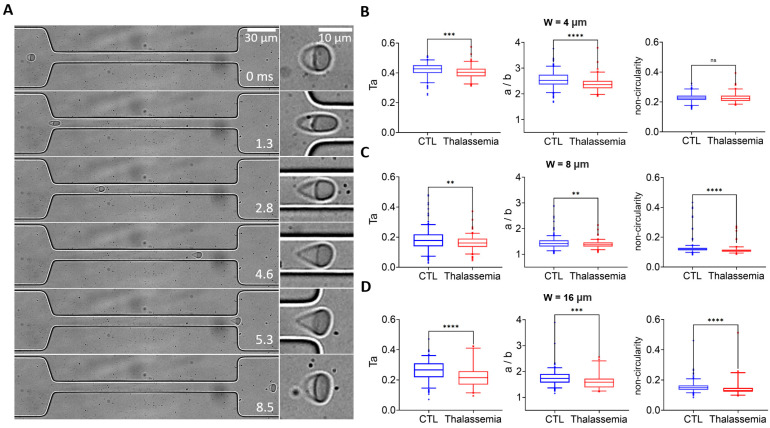
Deformability screening of RBCs in healthy and pathological conditions of thalassemia. (**A**) Transit and deformation of an example RBC from the patient with thalassemia in the microfluidic channel. An enlarged picture of the cell is shown on the right. Comparison of the deformability of RBCs between healthy donors and patients with thalassemia measured from narrow channels with widths of (**B**) 4 μm, (**C**) 8 μm, and (**D**) 16 μm. Deformability is defined by *Ta*, aspect ratio *a/b*, and non-circularity. The flow rate is 0.03 μL/min. ** *p* < 0.01, *** *p* < 0.001, **** *p* < 0.0001, one-way ANOVA test followed by a post-hoc Tukey HSD multiple comparison test.

**Figure 7 micromachines-16-00957-f007:**
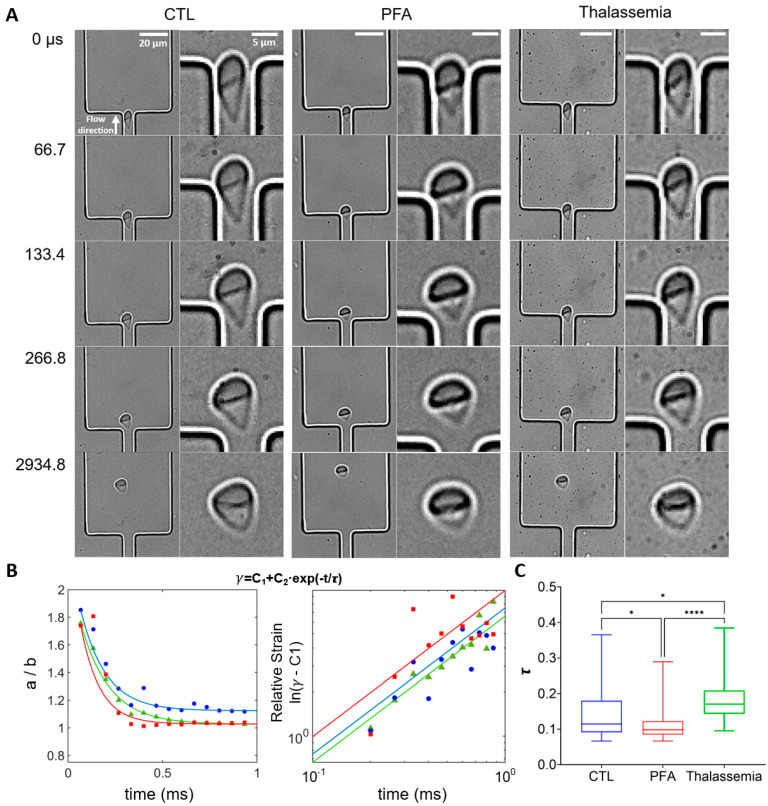
Analysis of shape recovery (strain decay) of RBCs from healthy control, PFA-treated healthy RBCs, and RBCs from patients with thalassemia. (**A**) Shape recovery of individual RBCs from different groups exiting the narrow stretching channel. The direction of flow is from bottom to top. The channel width is 8 μm, with V_n_ of 60 mm/s. (**B**) (**Left**) Representative strain γ (=*a/b*) for RBCs from healthy controls (blue), PFA-treated healthy RBCs (red), and RBCs from thalassemia patients (green) as a function of time. (**Right**) Double logarithmic plot of (γ − C1) as a function of time for the cells shown on the left. The solid lines represent the fits of the strain data to the exponential function γ = C1 + C2 × exp(−t/τ), where τ represents the characteristic relaxation time. τ for the cells are 0.15 s, 0.11 s, and 0.18 s for healthy RBCs, PFA-treated healthy RBCs, and RBCs from thalassemia patients, respectively. (**C**) Box plots of the characteristic relaxation time τ for the exponential decay fitting for each cell. The lines inside the boxes represent the median value of the data, while the edges of the boxes are the 25th and 75th percentiles. The goodness of the fit (average R2 value) is above 0.95 for all groups. *n* = 24, 47, and 48 for healthy controls (CTL), PFA-treated healthy RBCs, and RBCs from thalassemia patients. * *p* < 0.05, **** *p* < 0.0001, one-way ANOVA test followed by a post-hoc Tukey HSD multiple comparison test.

## Data Availability

Data are contained within the article.

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
