# Peer review of "A Microfluidic Approach for Assessing the Rheological Properties of Healthy Versus Thalassemic Red Blood Cells"

_micromachines, 2025, doi:10.3390/mi16080957_

Round 1

Reviewer 1 Report

Comments and Suggestions for Authors

The manuscript entitled "A microfluidic approach to assess the rheological properties of red blood cells in health and thalassemia” by Hao Jiang et al. describes the characterization of red blood cell (RBC) rheological properties using shear flow deformability cytometry (sDC). The manuscript is well-organized and provides new insights into the rheological properties of RBCs from patients with thalassemia, although the microfluidic device and the overall approach for assessing RBC mechanical properties lacked originality.

If the authors address the following comments, the manuscript could be considered for acceptance in “Micromachines”.

  1. Please include information on the image resolution of the high-speed camera. The number of pixels per single RBC is likely critical for tracking cells with the custom-built software and for analyzing cell contours using the deep learning method. In particular, the reviewer suggests adding a brief discussion on how the number of pixels per RBC influences the accuracy of cell deformation analysis.
  2. Please verify the time scales in Fig. 2A and the x-axis of Fig. 2C. In Figure 2A, the RBC appears to reach maximum deformation at approximately 1.7 msec, whereas the aspect ratio (a/b), representing cell deformation, seems to peak around 3.5 msec in Figure 2C. This discrepancy should be clarified or corrected.
  3. Please expand the discussion on how physical properties of cells, such as Young’s modulus, could be estimated using the current microfluidic platform. For example, the authors defined the parameter “n” as an index reflecting cell elasticity. Discussing the relationship between “n” and established mechanical properties like Young’s modulus would enhance the versatility and interpretability of the proposed method.
  4. Isn't Figure 2C on line 279 a mistake for Figure 3C? Please check.

Author Response

  1. Please include information on the image resolution of the high-speed camera. The number of pixels per single RBC is likely critical for tracking cells with the custom-built software and for analyzing cell contours using the deep learning method. In particular, the reviewer suggests adding a brief discussion on how the number of pixels per RBC influences the accuracy of cell deformation analysis.

We thank the reviewer for the valuable suggestion to improve our manuscript. Our high-speed camera has a resolution of 1028×1028 pixels, and we used a 63× objective lens. It is the highest magnification achievable under our current experimental conditions since we need to capture the whole motion trajectory of RBCs within the microfluidic channel and further magnification would reduce the field of view.

We agree that the number of pixels per single RBC is critical for tracking cells and analyzing cell contours. In our study, each red blood cell occupies approximately 510 pixels in total. We found it is sufficient to ensure accuracy when measuring cellular contours. We thank the reviewer for pointing this out. Now we have added this information to our main text:

“The number of pixels per single RBC is critical for tracking cells and analyzing cell contours … which we found was sufficient to ensure accuracy of cellular contour detection.”

  1. Please verify the time scales in Fig. 3A and the x-axis of Fig. 3C. In Figure 3A, the RBC appears to reach maximum deformation at approximately 1.7 msec, whereas the aspect ratio (a/b), representing cell deformation, seems to peak around 3.5 msec in Figure 2C. This discrepancy should be clarified or corrected.

We apologize for the mistake. We set t=0 when the velocity gradient starts developing near the constriction entrance in Fig. 3A but made mistake in Fig. 3C for the timing and traces (the previous traces were from another recording rather than that shown in panel A). We have now synchronized the time scales in both panels and updated Fig. 3 and its figure caption accordingly.

  1. Please expand the discussion on how physical properties of cells, such as Young’s modulus, could be estimated using the current microfluidic platform. For example, the authors defined the parameter “n” as an index reflecting cell elasticity. Discussing the relationship between “n” and established mechanical properties like Young’s modulus would enhance the versatility and interpretability of the proposed method.

We thank the reviewer for pointing this out. Previous studies have shown that the elastic shear modulus and viscous modulus of RBCs can be calculated in a similar microfluidic platform as we used (Reference 1-3 below). It was found that the average elastic shear modulus was higher for softer cells (Reference 1 below). Now we have discussed this method in our main text.

[1] D. C. Williams and D. K. Wood, High-throughput quantification of red blood cell deformability and oxygen saturation to probe mechanisms of sickle cell disease, PNAS 2023 Vol. 120 No. 48 e2313755120.

[2] R. M. Hochmuth, P. R. Worthy, E. A. Evans, Red cell extensional recovery and the determination of membrane viscosity. Biophys. J. 26, 101–114 (1979).

[3] J. E. Mancuso, W. D. Ristenpart, Stretching of red blood cells at high strain rates. Phys. Rev. Fluids 2, 101101 (2017).

  1. Isn't Figure 2C on line 279 a mistake for Figure 3C? Please check.

Yes, we apologize for this mistake and have corrected it accordingly.

Reviewer 2 Report

Comments and Suggestions for Authors

In this work the authors present the development of a high throughput experimental microfluidic-based method to evaluate the flow-induced deformation of red blood cells (RBC) as they are forced through a narrow channel.  To validate their method, they present results of (a) healthy RBC, (b) after treatment with paraformaldehyde (PFA) and (c) in the presence of thalassemia.  They show that the effects intensify when the channel is narrower and the flow stronger.  They follow three different measures of deformation: aspect ratio of a fitted ellipse, Taylor deformation, and deformability based on the area A and the perimeter P of the cells.  They showed decreased deformation in thalassemia as well as PFA treated cells as opposed to healthy cells.  They also followed the recovery of the shape of the cells after deformation and showed that the thalassemia cells take longer to recover.

The evaluation of the deformability of red blood cells is of medical importance since, for example, it can be associated to blood diseases.  It has therefore been the subject of many studies, several involving shear-induced deformation as discussed in this work and several applied to thalassemia, the illustrative example used in the present work. The novelty appears to be in streamlining the methodology, taking further advantage of specialized algorithms and machine-learning for cell-tracking, contour evaluation and deformation analysis as well as involving space and time-dependent analysis usually lacking in similar investigations.  I  do however have a number of questions, as described in the detailed comments below, that the authors need to respond satisfactorily in a recommended revision.

Detailed comments

  1. In presenting the literature of high-throughput microfluidic-based RBC deformation processes (lines 73-107), some relevant reviews can be mentioned (see, for example, references [1-3] supplied below).
  2. In assessing the importance of the study of RBC deformation (lines 108-110) several additional pathological conditions need to be mentioned, most certainly malaria (see, for example, reference [1] provided below and references therein) and diabetes (see, for example, reference [4] and references therein).
  3. In assessing work on thalassemia (lines 108-121), perhaps the following references can also be cited (references [5-6] supplied below).
  4. The information on the microfluidic channels used is not complete: The widths used and their (common?) length is offered, but not the depth of the channel. Related to that, the authors sometimes report the velocity (see, for example, lines 259, 261) sometimes the average flow rate (see, for example, lines 318 and 328).  Would be nice if there is an expression correlating the two that is presumably used in extracting the first (this is average velocity, right?) from the latter.  For that some assumptions need to be made on the rheology (presumably Newtonian?) and a numerical or approximate (using hydraulic radius for the channel and a Poiseuille expression?).  These need to be supplied (and justify the need for the supply of the information on the channel depth) . Related to these remarks, the average and maximum shear stress are also mentioned in lines 325-329.  These require some information on viscosity and the flow kinematics or its approximation: can those be provided?
  5. Certain names characterizing algorithms are mentioned (like the Hungarian algorithm, line 193; the Canny algorithm, line 198; the YOLOv8 convolutional neural network, line 203) but no references are supplied.
  6. In characterizing the RBC deformation the use of ellipses is made fit to the projected images of cell contours (line 221). However, RBCs have disc shape (as also mentioned in line 37) which may get even more complicated under deformation and is further distorted when projected in imaging.  Have the authors made any attempt to rationalize those effects in the data?  Any discussion along those lines (and relevant references if available) can be really helpful in physically interpreting the results.
  7. In several places (see, for example, lines 261 and lines 283-284) shear flow deformation is only considered as responsible for the RBC deformation. However, most clearly at the entrance and exit from the narrow channel we have a more complex flow with extensional deformation components.  Can those be considered as well in interpreting the results?
  8. In the analysis of time-dependent recovery an empirical equation is used (in line 378) involving a parameter indicated by “n” ; however, its similarity with a lower law exponent gives falsely the impression of a dimensionless coefficient (this is not true, n has units of inverse time). It is better if it is replaced by its inverse and denoted by τ this will necessarily have the units of time and can be better physically interpreted as a relaxation time of the medium.
  9. In the analysis of the results, as appeared in the discussion (lines 403 – 467), it may be of interest to see if the data can further be used in establishing information of the elastic and viscous moduli of RBC, following, for example, the analysis and formulae appearing in reference [7] provided below.
  10. Editorial remarks:
    1. Line 24: PFA needs to be spelled out
    2. Line 148: Together with providing the number of cells per volume, will be helpful to lso cite the corresponding concentration encountered in real blood (4,700 – 6,100) in same units to assess the dilution of the suspension.
    3. Line 226: “Taylor” is missplelled
    4. Lines 318, 327, 339, 345: uL should be μL.
    5. In References [24], [58], [59] and [61], capitalize author names

References

[1] Y. Chen, K. Guo, L. Jiang, S. Zhu, Z. Ni, N. Xiang, Microfluidic deformability cytometry: a review, Talanta 251 (2022) 123815.

[2] An L, Ji F, Zha.o E, Liu Y and Liu Y (2023), Measuring cell deformation by microfluidics. Front. Bioeng. Biotechnol. 11:1214544.

[3]  L. Fajdiga, S. Zemljic, T Kokalj, J. Derganc, Shear flow deformability cytometry: A microfluidic method advancing towards clinical use - A review, Analytica Chimica Acta 1355 (2025) 343894.

[4]  G. Eluru, R. Srinivasan, and S. S. Gorthi, Deformability Measurement of Single-Cells at High-Throughput With Imaging Flow Cytometry, Journal of Lightwave Technology, 33:3475-3480.

[5]  M. Sinan, O. Yalcin, Z. Karakas, E. Goksel, and N. Z. Ertana, Zinc improved erythrocyte deformability and aggregation in patients with beta-thalassemia: An in vitro study, Clinical Hemorheology and Microcirculation 85 (2023) 1–12.

[6]  Krishnevskaya, E.; Molero, M.; Ancochea, Á.; Hernández, I.; Vives-Corrons, J.-L. New-Generation Ektacytometry Study of Red Blood Cells in Different Hemoglobinopathies and Thalassemia. Thalass. Rep. 2023, 13, 70–76.

[7]  D. C. Williams and D. K. Wood, High-throughput quantification of red blood cell deformability and oxygen saturation to probe mechanisms of sickle cell disease, PNAS 2023 Vol. 120 No. 48 e2313755120.

Author Response

Detailed comments

  1. In presenting the literature of high-throughput microfluidic-based RBC deformation processes (lines 73-107), some relevant reviews can be mentioned (see, for example, references [1-3] supplied below).

We thank the reviewer for the helpful suggestion to improve our manuscript. The recommended references have now been added to the revised manuscript (Ref. 24-26):

“In contrast, microfluidic-based methods [23] represent a highly attractive alternative due to their potential for single-cell precision, high-throughput deformability analysis [24-26].”

2. In assessing the importance of the study of RBC deformation (lines 108-110) several additional pathological conditions need to be mentioned, most certainly malaria (see, for example, reference [1] provided below and references therein) and diabetes (see, for example, reference [4] and references therein).

References added accordingly: “The deformability of RBCs is influenced by many pathological conditions, such as malaria [45], diabetes [46, 47], sickle cell disease [48], thalassemia [49], hereditary spherocytosis[50] and hereditary xerocytosis [51]”.

3. In assessing work on thalassemia (lines 108-121), perhaps the following references can also be cited (references [5-6] supplied below).

We have added reference [6] recommended by the reviewer:

“Caused by autosomal gene defects, the disease results in an imbalance in the synthesis ratio of α- and β-globin chains in hemoglobin [54, 55]

We didn’t cite reference [5] as it is not closely related to our main text here and we have cited sufficient references for this part.

  1. The information on the microfluidic channels used is not complete: The widths used and their (common?) length is offered, but not the depth of the channel. Related to that, the authors sometimes report the velocity (see, for example, lines 259, 261) sometimes the average flow rate (see, for example, lines 318 and 328).  Would be nice if there is an expression correlating the two that is presumably used in extracting the first (this is average velocity, right?) from the latter.  For that some assumptions need to be made on the rheology (presumably Newtonian?) and a numerical or approximate (using hydraulic radius for the channel and a Poiseuille expression?).  These need to be supplied (and justify the need for the supply of the information on the channel depth) . Related to these remarks, the average and maximum shear stress are also mentioned in lines 325-329.  These require some information on viscosity and the flow kinematics or its approximation: can those be provided?

We thank the reviewer for the constructive comments to improve our manuscript. The depth of the narrow channels are kept constant as 7 µm. The average flow velocity  in the channel and the average flow rate Q is related by the following equation:

where S is the cross-sectional area of the channel and can be calculated as the channel width x channel depth. It is important to note that the velocities of the cells are calculated from their trajectory from experiment. Now we have stated this information more clearly in the main text.

As we used a highly diluted RBC suspension (10⁷ cells/mL in DPBS), the viscosity of the fluid is close to that of the buffer 1x DPBS and similar to that of water. The medium can be assumed as an incompressible Newtonian fluid. With ρ = 998 kg m⁻³ (density of water at room temperature), U = 0.01–0.1 m s⁻¹ (the characteristic flow velocity in our study), L = 1–10 µm (the characteristic length, taken as the channel width), and μ= 0.8937×10⁻³ Pa∙ s (dynamic viscosity of water), the Reynolds number  ranges from 0.01 to 1.12, well below the laminar–turbulent transition threshold of ~2000, confirming that the flow is laminar.

In our numerical simulation, we have assumed no-slip boundary condition at the channel wall and the flow do have a Poiseuille profile. As we used sheath flow and focused RBCs to the channel center, we defined that the average flow shear stress along the flow direction at the entrance of the narrow channel and along the center line can be calculated as ?ave=3μ ∆u/∆x [Ref.1], where μ is the dynamic viscosity of the medium, ∆u/∆x is the average velocity gradient along the flow direction (along the centerline of the channel).

The max flow shear stress along the center streamline is calculated from the maximum value of the first-order derivative of the position dependent velocity curves in Fig. 5D (?max=3μ (du/dx)max), as shown by the peaks of the curves in Fig.5F and Fig. 5G

We have incorporated this information and improved the main text accordingly.

Ref.1: Mancuso J E, Ristenpart W D. Stretching of red blood cells at high strain rates[J]. Physical Review Fluids, 2017, 2(10): 101101.

5. Certain names characterizing algorithms are mentioned (like the Hungarian algorithm, line 193; the Canny algorithm, line 198; the YOLOv8 convolutional neural network, line 203) but no references are supplied.

Thanks for pointing this out. References have now been added accordingly.

6. In characterizing the RBC deformation the use of ellipses is made fit to the projected images of cell contours (line 221). However, RBCs have disc shape (as also mentioned in line 37) which may get even more complicated under deformation and is further distorted when projected in imaging.  Have the authors made any attempt to rationalize those effects in the data?  Any discussion along those lines (and relevant references if available) can be really helpful in physically interpreting the results.

Thank the reviewer for the constructive comments.

RBCs travelling through microfluidic constriction channel typically adopt a limited set of morphologies. We restricted our analysis to cells exhibiting the parachute shape, which is most common in our study and reported by other previous studies as well in similar microfluidic platform.

We were able to detect the cell contour accurately, obtained the projected cell area A and perimeter P, and used four definitions of cell deformation. We only assume an ellipse fit to the cell contour for aspect ratio and Taylor deformation by getting the major and minor axis a and b. For deformability (non-circularity index) and changes in the major axis, we didn’t fit the cell contour to an ellipse.

Now we have discussed these in “section 2.5.3. Cell deformation analysis”.

7. In several places (see, for example, lines 261 and lines 283-284) shear flow deformation is only considered as responsible for the RBC deformation. However, most clearly at the entrance and exit from the narrow channel we have a more complex flow with extensional deformation components.  Can those be considered as well in interpreting the results?

Based on both experimental recordings and numerical simulations, the steep velocity gradients at the entrance and exit of the narrow channel expose RBCs to markedly higher shear stress, resulting in greater deformation.

At lines 261 and lines 283-284, we are indeed describing the velocity gradient along the flow direction at the entrance and exit along the channel centerline, which impose extensional deformation. The shear stress we described and calculated are indeed related to this velocity gradient. Now we have made it clearer in the main text:

“Based on both experimental recordings and numerical simulations, the steep velocity gradients along the flow direction (x) and the channel centerline at the entrance and exit of the narrow channel expose RBCs to markedly higher shear stress, resulting in notable extensional deformation. The shear stress ? herein refers to that related to this velocity gradient.”

8. In the analysis of time-dependent recovery an empirical equation is used (in line 378) involving a parameter indicated by “n”; however, its similarity with a lower law exponent gives falsely the impression of a dimensionless coefficient (this is not true, n has units of inverse time). It is better if it is replaced by its inverse and denoted by τ this will necessarily have the units of time and can be better physically interpreted as a relaxation time of the medium.

We thank the reviewer for the insightful suggestion to improve our manuscript. We agree that n has units of inverse time, and it is better to replace it with its inverse τ. We have updated Fig.7 and its figure caption and the text accordingly in the manuscript.

9. In the analysis of the results, as appeared in the discussion (lines 403 – 467), it may be of interest to see if the data can further be used in establishing information of the elastic and viscous moduli of RBC, following, for example, the analysis and formulae appearing in reference [7] provided below.

We thank the reviewer for the insightful suggestions. We learned from reference [7] and other relevant references and calculated the elastic shear modulus and viscous modulus of representative RBCs from the control and thalassemia groups (with their shape recovery speed (e.g., now the 1/?) at around the mean value of their respective groups). Now we have mentioned the methods and results in “2.5.4. Cell shape recovery analysis” and the discussion section.

[7] D. C. Williams and D. K. Wood, High-throughput quantification of red blood cell deformability and oxygen saturation to probe mechanisms of sickle cell disease, PNAS 2023 Vol. 120 No. 48 e2313755120.

10. Editorial remarks:

    1. Line 24: PFA needs to be spelled out
    2. Line 148: Together with providing the number of cells per volume, will be helpful to lso cite the corresponding concentration encountered in real blood (4,700 – 6,100) in same units to assess the dilution of the suspension.
    3. Line 226: “Taylor” is missplelled
    4. Lines 318, 327, 339, 345: uL should be μL.
    5. In References [24], [58], [59] and [61], capitalize author names

Thanks for pointing these out. Changed accordingly.

References

[1] Y. Chen, K. Guo, L. Jiang, S. Zhu, Z. Ni, N. Xiang, Microfluidic deformability cytometry: a review, Talanta 251 (2022) 123815.

[2] An L, Ji F, Zha.o E, Liu Y and Liu Y (2023), Measuring cell deformation by microfluidics. Front. Bioeng. Biotechnol. 11:1214544.

[3]  L. Fajdiga, S. Zemljic, T Kokalj, J. Derganc, Shear flow deformability cytometry: A microfluidic method advancing towards clinical use - A review, Analytica Chimica Acta 1355 (2025) 343894.

[4]  G. Eluru, R. Srinivasan, and S. S. Gorthi, Deformability Measurement of Single-Cells at High-Throughput With Imaging Flow Cytometry, Journal of Lightwave Technology, 33:3475-3480.

[5]  M. Sinan, O. Yalcin, Z. Karakas, E. Goksel, and N. Z. Ertana, Zinc improved erythrocyte deformability and aggregation in patients with beta-thalassemia: An in vitro study, Clinical Hemorheology and Microcirculation 85 (2023) 1–12.

[6]  Krishnevskaya, E.; Molero, M.; Ancochea, Á.; Hernández, I.; Vives-Corrons, J.-L. New-Generation Ektacytometry Study of Red Blood Cells in Different Hemoglobinopathies and Thalassemia. Thalass. Rep. 2023, 13, 70–76.

[7]  D. C. Williams and D. K. Wood, High-throughput quantification of red blood cell deformability and oxygen saturation to probe mechanisms of sickle cell disease, PNAS 2023 Vol. 120 No. 48 e2313755120.

Reviewer 3 Report

Comments and Suggestions for Authors

The authors investigate a question of scientific and practical importance. The Introduction presents a clear, well-organized overview of prior research, properly citing relevant work. Methods are detailed sufficiently. Results are presented clearly and appropriately tabulated. The Discussion thoughtfully interprets the results in the context of existing literature. The manuscript is well-written and likely to appeal to a broad audience. However, it would benefit from a more critical discussion of the experimental limitations and boundary conditions.

Recommendation: Accept with revisions.

Revisions needed:

Add a dedicated paragraph on limitations (in the Discussion).

Limitations section:

  1. Address the physiological relevance of the used conditions (such as the applied level of shear stress, room temperature, and the use of a buffer). Explain how these factors might affect extrapolation to in vivo conditions.
  2. As indicated by the authors (see the legend to Fig. 4), the number of analyzed cells did not exceed 150. At the same time, the data show considerable variability in the measured parameters across the cell population. In light of this, the authors should discuss the robustness of their conclusions, considering both the relatively small sample size (fewer than 150 cells) and the substantial variability observed in the measurements.

These revisions will help clarify the experimental context and provide deeper insights into interpretation.

Author Response

Add a dedicated paragraph on limitations (in the Discussion).

Limitations section:

  1. Address the physiological relevance of the used conditions (such as the applied level of shear stress, room temperature, and the use of a buffer). Explain how these factors might affect extrapolation to in vivo conditions.
  2. As indicated by the authors (see the legend to Fig. 4), the number of analyzed cells did not exceed 150. At the same time, the data show considerable variability in the measured parameters across the cell population. In light of this, the authors should discuss the robustness of their conclusions, considering both the relatively small sample size (fewer than 150 cells) and the substantial variability observed in the measurements.

These revisions will help clarify the experimental context and provide deeper insights into interpretation.

We thank the reviewer for the constructive comments to improve our manuscript. A paragraph on limitations has been added to the Discussion section:

“To approximate physiological conditions, the dimensions of the microfluidic constriction channel (7 µm in depth and 4, 8, 16 µm in width), the resulting RBC velocities(on the order of 10-100 mm/s)and shear stress for extensional deformation (1-102 dynes/cm2) fall within the ranges reported for human blood vessels and flow speed of human erythrocytes[80, 81]. However, several limitations exist. First, the microchannel is rectangular shaped in our study instead of cylindrical geometry as in vivo. Therefore, RBCs were exposed to a more heterogenous shear stress in the experiment compared to homogeneous shear stress in the radial direction in vivo. Nevertheless, we are more interested in the velocity gradient in the flow direction along the channel centerline and the resulting shear stress. Second, we used a highly diluted and RBC-only suspension in DPBS buffer prepared from whole blood. Thus, the extracellular environment differs markedly from native blood plasma, e.g., the fluid viscosity/density, chemical environment and cell-cell communications could differ significantly [82]. Third, our experiment was performed at room temperature, which is lower than that in vivo and may affect the viscoelastic properties of RBCs and their deformability [83]. Fourth, our studies have a limited sample size (<150 cells) and considerable variability is observed in the measured parameters across the cell population. Due to the large field of view and high frame rate required to capture complete cell trajectories and deformations, tens of thousands of high-resolution images were generated within seconds. This resulted in a limitation on the number of cells that could be recorded per unit time, owing to the constrained storage capacity of our high-speed imaging system. The variability in the measured parameters could arise from the inherent heterogeneity within the RBC population [84]. Nevertheless, by employing sheath flow to focus cells toward the channel center and precisely detecting their contours, we reliably measured and distinguished statistically significant differences in cell deformability across groups.”